# Predicting N6-Methyladenosine Sites in Multiple Tissues of Mammals through Ensemble Deep Learning

**DOI:** 10.3390/ijms232415490

**Published:** 2022-12-07

**Authors:** Zhengtao Luo, Liliang Lou, Wangren Qiu, Zhaochun Xu, Xuan Xiao

**Affiliations:** Computer Department, Jingdezhen Ceramic University, Jingdezhen 333403, China

**Keywords:** RNA modification, m^6^A site identification, ensemble deep learning

## Abstract

N6-methyladenosine (m^6^A) is the most abundant within eukaryotic messenger RNA modification, which plays an essential regulatory role in the control of cellular functions and gene expression. However, it remains an outstanding challenge to detect mRNA m^6^A transcriptome-wide at base resolution via experimental approaches, which are generally time-consuming and expensive. Developing computational methods is a good strategy for accurate in silico detection of m^6^A modification sites from the large amount of RNA sequence data. Unfortunately, the existing computational models are usually only for m^6^A site prediction in a single species, without considering the tissue level of species, while most of them are constructed based on low-confidence level data generated by an m^6^A antibody immunoprecipitation (IP)-based sequencing method, thereby restricting reliability and generalizability of proposed models. Here, we review recent advances in computational prediction of m^6^A sites and construct a new computational approach named im6APred using ensemble deep learning to accurately identify m^6^A sites based on high-confidence level data in multiple tissues of mammals. Our model im6APred builds upon a comprehensive evaluation of multiple classification methods, including four traditional classification algorithms and three deep learning methods and their ensembles. The optimal base–classifier combinations are then chosen by five-fold cross-validation test to achieve an effective stacked model. Our model im6APred can produce the area under the receiver operating characteristic curve (AUROC) in the range of 0.82–0.91 on independent tests, indicating that our model has the ability to learn general methylation rules on RNA bases and generalize to m^6^A transcriptome-wide identification. Moreover, AUROCs in the range of 0.77–0.96 were achieved using cross-species/tissues validation on the benchmark dataset, demonstrating differences in predictive performance at the tissue level and the need for constructing tissue-specific models for m^6^A site prediction.

## 1. Introduction

Chemical modification is an efficient and important way to regulate the functions of biological macromolecules, including protein, DNA and RNA [1]. Besides modifications of protein and DNA, more than 150 different RNA post-transcriptional modifications have been characterized so far [2]. Among them, N6-methyladenosine (m^6^A), the methylation of adenosine at position 6, is the most abundant and prevalent on RNA molecules present in eukaryotes [3], regulated mainly by a variety of “writer”, “reader” and “eraser” proteins [4] (Figure 1). This modification has been proved to be implicated in a variety of cellular functions, such as the self-renewal and differentiation of stem cells [5], DNA-damage response [6], spermatogonia differentiation [7], cellular heat shock response [8], anti-tumor immunity [9], X-chromosome inactivation [10], long-term memory creation [11], circadian clock function [12] and tumorigenesis [13]. In addition, aberrant m^6^A methylation is also closely related to the disease progression, including acute myeloid leukemia, breast tumor, gastric cancer and glioblastoma. Therefore, identifying m^6^A modification sites accurately is crucial to understand and explore the regulatory mechanisms and functions of various RNAs.

The m^6^A modification is difficult to be discriminated by chemical reactions because its chemical properties are similar to adenosine. Thus, to accurately profile the transcriptome-wide distribution of m^6^A modification, a number of high-throughput m^6^A sequencing techniques have been developed, including MeRIP [14] or m^6^A-seq [15], PA-m^6^A-seq [16], miCLIP [17], UV-CLIP [18], m^6^A-REF-seq [19] and DART-seq [20]. Despite significant progress achieved in transcriptome-wide mapping of m^6^A at the nucleotide level, these experimental methods are expensive and time-consuming for genome-scale detection of m^6^A modification sites. Therefore, it is urgent to develop computational methods to identify m^6^A modification sites in RNA as an effective bioinformatics tool for scholars to study RNA m^6^A biology.

Recent advances in high-throughput m^6^A sequencing techniques and the accumulation of experimentally validated m^6^A modification data [21,22,23] have paved the way for training such in silico predictors. To date, numerous computational tools have been developed for in silico prediction of m^6^A modification sites from RNA sequences, including iRNA-Methyl [24], M6ATH [25], MethyRNA [26], iRNA-PseColl [27], DeepM6APred [28], iRNA(m6A)-PseDNC [29], RAM-ESVM [30], RAM-NPPS [31], RNAMethPre [32], iRNA-3typeA [33], M6APred-EL [34], pRNAm-PC [35], TargetM6A [36], AthMethPre [37], iMethyl-STTNC [38], SRAMP [39], HMpre [40], MASS [41], MultiRM [42], etc. A summary of the existing methods for m^6^A modification site prediction are shown in Table 1.

Although promising achievements have been made in this field, as shown in Table 1, existing approaches are subject to the following limitations. Firstly, most of them were developed using traditional machine learning methods, such as support vector machine (SVM) [24,25,26,27,28,29,30,31,32,33,34,35,36,37,38,43], random forest (RF) [39] and eXtreme gradient boosting (XGBoost) [40]. However, few computational tools [41,42,44,45] were developed based on deep learning, which has powerful capacity to extract meaningful feature representations from a variety of raw data. Secondly, most existing studies relied on a limited number of m^6^A modification data from a single species, such as *S. cerevisiae*, *A. thaliana* and *H. sapiens*. While few computational predictors were proposed for identifying m^6^A modification sites from multiple species, especially in different tissues. Thirdly, the benchmark data of most existing tools in the field consisted of m^6^A modification sites with a medium confidence level. These sites were derived from m^6^A-seq, which cannot identify precise m^6^A modification sites and can only provide a 100–200 nt mapping region with m^6^A. More high confidence m^6^A modification sites are required to be directly extracted from the experiments with the single-nucleotide level, making the prediction results more reliable.

Recently, Dao et al. [46] collected experimentally confirmed single-nucleotide resolution m^6^A sites in various tissues of humans, mice and rats to construct a high-quality dataset and then proposed an SVM-based predictor named iRNA-m6A for identifying m^6^A sites. Although the performance of iRNA-m6A is promising for m^6^A site prediction in various tissues of multiple species, the classification method that they adopted focused on traditional machine learning methods, restricting further improvements of performance. Subsequently, to further improve the performance for m^6^A site prediction at the tissue level, Liu et al. [47] used the one-hot encoding scheme and a convolutional neural network (CNN) containing a convolutional layer and a max-pooling layer to propose a new model called im6A-TS-CNN based on Dao’s benchmark datasets. Abbas et al. [48] proposed a CNN model TS-m6A-DL containing three convolutional and max-pooling blocks, where the corresponding flattened outputs of each block were concatenated and fed into the dense layer.

Ensemble learning and deep learning have achieved great achievements in bioinformatics because of their superior adaptability and flexibility. However, these two methods were mainly treated as two kinds of independent methodologies. Recently, their combination formed a new methodology, termed ensemble deep learning, for improving stability and generalization capability of the proposed model synergistically. Compared with one single deep learning model, ensemble deep learning has been confirmed to pose better generalizability, prompting a new wave of research and application in different bioinformatics fields, such as multi-omics [49,50,51], systems biology [52,53,54], structural bioinformatics [55,56], sequence analysis [57] and so on.

Motivated by these remarkable achievements of ensemble deep learning, we here propose an ensemble deep-learning-based method, named im6APred, to identify m^6^A modification sites from the primary RNA sequence in various tissues of multiple species, including (a) human brain, liver and kidney, (b) mouse brain, liver kidney, heart, and testis, and (c) rat brain, liver and kidney. The predictor im6APred mainly consists of two core components: (i) five base models that predict m^6^A modification sites; and (ii) a simple ensemble method for achieving more stable results. As diversity of an individual network is a critical factor of a good ensemble model, the base models were generated based on convolutional neural network (CNN) under different hyperparameter combinations for promoting diversity of the individual network. Then, an ensemble strategy that averages output from all the individual base models was adopted to combine these base models into a final ensemble model. Overall workflow of the proposed predictor im6APred is illustrated in Figure 2. In addition, we developed a user-friendly webserver, which is publicly accessible at http://47.94.248.117/im6APred (accessed on 3 December 2022) [58].

## 2. Results and Discussion

### 2.1. The Optimal Base-Classifier Combinations

An effective ensemble learning strategy can achieve the construction of a robust prediction model by integrating the information of various classifiers. In this study, we employed four commonly used machine learning algorithms and three deep learning methods as base classifiers, including support vector machine (SVM), k-nearest neighbors (KNN), random forest (RF), gradient boosting (GB), fully connected network (FCN), long short-term memory (LSTM) and convolutional neural network (CNN). To elucidate advantages of ensemble learning, we firstly evaluated predictive performance of each single classifier trained on specific tissue data of each species using five-fold validation. Their optimal parameters are determined by the grid search method during five-fold validation. This process can be implement using Grid Search CV in python, which tries all the exhaustive combinations of parameter values supplied by a user and selects the best from the parameter space. The optimal parameter combinations of these classifiers were deposited in Appendix A. Subsequently, we selected five different single classifiers among them according to the ACC metric as the base classifiers and then produced 21 base–classifier combinations. In addition, we also considered seven other base–classifier combinations generated by the same kind of single classifiers. In detail, in accordance with the ACC metric, we selected the five best performing models from the same kind of single classifiers with different hyper-parameters. Finally, a simple average method was employed as the ensemble strategy of these 35 base–classifier combinations to classify.

We comprehensively evaluated these 35 base–classifier combinations for human brain tissue and other various tissues of multiple species and listed the five-fold cross-validation results in Table 2 and Appendix A, respectively. As can be seen, for a single classifier, CNN and LSTM models have better performance than other classifiers, and so do the ensemble models containing CNN and LSTM, indicating that CNN and LSTM could capture latent information of sequential features of RNA m^6^A modification by integrating possible local-range dependencies and long-range dependencies, respectively. Moreover, it is clear that the ensemble deep learning model generated by five CNNs with different hyper-parameters achieved the best performance in terms of almost all performance metrics, with the only exception of Sp. Therefore, we developed 11 m^6^A site predictors, collectively called im6APred, which integrate an ensemble of 5 CNN classifiers with the combined strategy of a simple average method for predicting m^6^A site from various tissues of multiple species.

### 2.2. Performance Evaluation of the Proposed Model at Tissue Level of Species

There are several published methods for predicting N6-methyladenosine sites in multiple tissues of mammals, such as iRNA-m6A, m6A-TS-CNN and TS-m6A-DL. To provide a fair comparison, we adopted the same assessment criteria to examine the existing methods. Five-fold cross-validation was used to evaluate performance of the proposed models, which were trained on training data from each tissue of the different species are shown in Table 1. Subsequently, the independent datasets of the different tissues from three species shown in Table 1 were used to evaluate the robustness and reliability of the corresponding model, respectively. The results from five-fold cross-validation and an independent test of our proposed models are all listed in Table 3. As shown in Figure 3, some slight differences between the AUROC values of five-fold cross-validation and the independent test indicate that the proposed method in the current study is robust.

To highlight the generalization ability of our tissue-specific models for predicting m^6^A modification sites, the corresponding results from five-fold cross-validation and an independent test obtained by their rivals are listed in Appendix A, respectively. The results from the independent test show that our proposed model im6APred has overall higher performance than other existing models. As shown in Figure 4, the values of the most important metrics, MCC and AUROC, are increased to a certain degree. For example, compared with iRNA-m6A, our predictor for identifying m^6^A sites in mouse liver obtained the maximum AUROC growth of 0.12 and increased 0.09 compared to the other two predictors. It indicates that im6APred has good stability and high generalization ability for identifying the m^6^A modification sites for unknown RNA sequences.

To verify the superior generalization ability of our tissue-specific model in comparison with existing tools, we constructed three other datasets, named HEK293_data, HEK293T_data and HepG2_data, respectively, from the recent literature created by Song et al. [59]. Among them, HEK293_data and HEK293T_data originated from human kidney, while HepG2_data is from human liver. We eliminated overlapping sequences from the training data so that the constructed data had not been seen in the training data. The ratios of positive and negative samples were all set to 1:1. After removing sequence redundancy, the sizes of such datasets were as follows: 8440 for HEK293_data, 29740 for HEK293T_data, and 8574 for HepG2_data. These datasets can be downloaded from http://47.94.248.117/im6APred/download (accessed on 3 December 2022).

The results on the independent test are listed in Table 4, and the corresponding ROC curves are illustrated in Figure 5. It can be seen that m6A-TS-CNN achieved better generalization ability than the other two existing predictors on these three independent testing datasets, while im6APred showed the best predictive performance among all the tissue-specific models. More specifically, the ACC, MCC and AUROC for the im6APred model outperformed TS-m6A-DL on the independent test dataset HEK293_data by 3.91%, 7.58% and 2.36%, respectively. The performance of the im6APred model on HEK293_data was improved compared to m6A-TS-CNN by 2.04%, 3.98% and 1.96% in terms of ACC, MCC and AUROC, respectively. Moreover, those values of ACC, MCC and AUROC were increased compared to iRNA-m6A by 3.56%, 7.15% and 1.14%, respectively. In addition, the ACC, MCC and AUROC for the im6APred model outperformed m6A-TS-CNN on the independent test dataset HEK293T_data by 3.58%, 6.81% and 3.52%, respectively. The performance of the im6APred model on HepG2_data was improved compared to m6A-TS-CNN by 1.91%, 3.7% and 3.01% in terms of ACC, MCC and AUROC, respectively.

### 2.3. Cross-Species/Tissues Validation

Since the benchmark datasets consist of various tissues of multiple species, it is necessary to further investigate the prediction capability of the proposed model on cross-species/cross-tissues data. Here, we implemented cross-species/cross-tissues validation experiments, in which the eleven tissue-specific models were trained on the training data of each tissue from three species, respectively. Subsequently, the remaining different tissue data were considered as independent test data to assess the performance of each tissue-specific model.

A heat map generated by the AUROCs is shown in Figure 6 to describe the prediction performance of tissue-specific models. Results in Figure 6 show that almost all cross-tissue prediction performance is accepted as all the AUROCs are higher than 0.8 in this heat map. Especially, the tissue-specific models trained on the datasets of human (liver and kidney), mouse (brain and kidney) and rat (brain, liver and kidney), respectively, have achieved superior results (AUROCs > 0.85), indicating that potential m^6^A sites in the sequences of different tissue can be accurately identified using these tissue-specific models. However, there is tissue specificity between human (brain) and mouse (liver, heart and testis), as confirmed by Dao et al. [46]. When testing the tissue-specific models trained on the human (brain) and mouse (liver), almost all the AUROC values produced were below 0.82, demonstrating that tissue-specific predictors are helpful for better detection of m^6^A sites from a specific tissue.

### 2.4. Weberver

To facilitate community-wide efforts in quick prediction of novel potential RNA m^6^A modification sites from RNA sequence data, we developed a user-friendly webserver of im6APred, which is publicly accessible at http://47.94.248.117/im6APred (accessed on 3 December 2022). The webserver im6APred can realize the prediction function for m^6^A modification sites at the species level or at the tissue level of species after specifying the tissue and species type in the drop-down menu simultaneously and inputting their sequences of interest or uploading an input sequence file in the FASTA format. Upon completion, the final prediction results will be displayed on the webpage or sent to the e-mail address users specified. A detailed user guide about how to use the webserver of im6APred can be found on the help page. In addition, all the data and the source code used in this study are provided on this webserver, which facilitates a further in-depth analysis of m^6^A modification.

## 3. Materials and Methods

### 3.1. Benchmark Dataset

Comprehensive interrogating of m^6^A at individual nucleotide resolution plays a crucial role in revealing the biological importance of this RNA modification. Most importantly, high-throughput m^6^A sequencing data with a single-nucleotide level is a pivotal resource as high-quality data for in silico identification of m^6^A modification sites accurately. However, most of the current computational methods can only predict m^6^A sites on a sample level. Recently, Zhang et al. [19] reported an antibody-independent enzymatic method termed m^6^A-REF-seq and identified a great number of m^6^A modification sites with the RNA ACA motif in different tissues of humans, mice and rats at a single-nucleotide level. These data can pave the way for the development of such computational predictors for predicting m^6^A sites in multiple tissues of mammals. According to the genome of humans (hg19), mice (mm10) and rats (rn6) downloaded from the University of California, Santa Cruz (UCSC), we employed a sliding window method with 41 nt length to construct the benchmark datasets based on such m^6^A site information of each species, including chromosome type, site position and strand. The sequence fragment is regarded as a positive sample if its center adenine in ACA motif is methylated. Otherwise, it is regarded as a negative sample if its center adenine in ACA motif is un-methylated. The final datasets were generated after removing the sequences with more than 80% sequence similarity by using CD-HIT software (Version 4.6.8, Burnham Institute for Medical Research, La Jolla, CA, USA) [60]. In this study, we first divided the benchmark dataset for specific tissue of each species into two equal parts randomly, one as a training set and the another as an independent testing set (Table 5), and then separately performed the analysis. In the process of model training, 5-fold cross-validation was implemented on the training set to optimize the parameters. Under the optimal parameter combination, all the training samples for specific tissue of each species were used to train the best model, respectively, and then the performance of the constructed model was evaluated using the corresponding independent testing set. The detailed procedure of benchmark data construction also can be seen in Dao et al. [46]. Additionally, the corresponding code for constructing the benchmark dataset and the exact data have been released on the Github https://github.com/pythonLzt/im6APred (accessed on 3 December 2022) to allow better reproducibility of the research.

### 3.2. Sample Formulation

Better feature representations can be conducive to distinguishing m^6^A sites from non-m^6^A sites more accurately, thereby improving the prediction performance of the proposed model. In the current study, we integrated two kinds of feature representations to represent the sample sequences.

#### 3.2.1. Nucleotide Chemical Property

Nucleotide chemical property is an efficient and effective sequence encoding method, playing an important part in the prediction of other modification sites, such as 4 mC [61] and D sites [62]. The RNA sequences can be encoding by following the steps below.

Firstly, a three-dimensional vector ($x_{i},\;y_{i},\;z_{i}$), in which its components represent the ring structure, the hydrogen bond and the chemical functionality of the four nucleotides, respectively, is used to encode a nucleotide in RNA sequence, as shown in Equation (1). Thus, A, C, G and U can be encoded by (1, 1, 1), (0, 0, 1), (1, 0, 0) and (0, 1, 0), respectively.
(1)$$x_{i}=\{\begin{array}{c} 1\;when\;R_{i}\;\in \left\{A,\;G\right\}\\ 0\;when\;R_{i}\;\in \left\{C,\;U\right\} \end{array},\;y_{i}=\{\begin{array}{c} 1\;when\;R_{i}\;\in \left\{A,\;U\right\}\\ 0\;when\;R_{i}\;\in \left\{C,\;G\right\} \end{array},\;z_{i}=\{\begin{array}{c} 1\;when\;R_{i}\;\in \left\{A,\;C\right\}\\ 0\;when\;R_{i}\;\in \left\{G,\;U\right\} \end{array}$$
where $R_{i}$ represents the i-th nucleotide in the RNA sample sequence.

Next, the density of nucleotide $R_{i}$ is used to represent this nucleotide by calculating the accumulated frequency of nucleotides along the RNA sequence, as formulated by Equation (2).
(2)$$d_{i}=\frac{1}{i}\sum_{k=1}^{i}f\left(R_{k}\right),\;f\left(R_{k}\right)=\{\begin{array}{c} 1\;if\;R_{k}=R_{i}\\ 0\;otherwise \end{array}$$
where 1≤i≤L, L is the RNA sequence length.

The combination of nucleotide chemical property and nucleotide density can quantitatively represent the given RNA sample sequence to the most extent. Following this method, each RNA sample sequence with 41 nt length could be converted into a 4 × 41-dimensional vector.

#### 3.2.2. Mono-Nucleotide Binary Encoding

Mono-nucleotide binary encoding is another simple and efficient method to encode sample sequences, widely used in representing nucleotide sequences and transforming them into numeric vectors [63,64,65,66]. Generally, A, C, G and U can be transferred to a 4-dimensional vector: (1, 0, 0, 0), (0, 1, 0, 0), (0, 0, 1, 0) and (0, 0, 0, 1), respectively.

As these encoding methods only consider the nucleotides along either sense strand or non-sense strand, a lot of sequence order information could be lost. Here, to capture the long- and local-range sequence order information of RNA sequences as much as possible, these two encoding methods were employed to encode both sense and non-sense strands simultaneously. Thus, the given RNA sample sequence with 41 nt length in this study could be transferred into a 16 × 41-dimensional vector finally.

### 3.3. Classification Method

An ensemble across multiple models or within a single model with appropriate ensemble strategies could achieve complementary learning of the training data, thereby greatly improving model reliability, accuracy and efficiency. Moreover, convolutional neural network (CNN) is a multi-layer neural network framework, which has been widely applied in many bioinformatics fields and has made significant progress [67,68,69,70,71,72,73,74,75,76]. Thus, in the current study, we adopted a common ensemble method based on model perturbation, training five CNN base classifiers with different hyper-parameters and considering a simple average method as an ensemble strategy to classify (Figure 2).

Generally, a CNN is mainly composed of three parts: convolution layers, pooling layers and fully connected layers. A set of filters of the convolution layer slide over the output of the previous layer to extract high-level features and generate a multiple feature map. The pooling layer following each convolution layer is used for extracting dominant features from the feature map. Finally, the fully connected layer, which employs a dropout strategy for mitigating potential overfitting, maps the learned features to the label space for classifying and predicting. It is well-known that the filter size and the number of convolution kernels are main determinants of the performance of a CNN framework. Here, we denote CNN base classifiers with x convolution layer(s), y convolution kernel(s), filter size of z and drop out probability of w. In the training process, we select the best five models as base classifiers from different settings of parameters by five-fold cross-validation. In detail, we took x from {1, 2, 3, 4}, y from {4, 8, 16, 32, 64}, z from {2, 4} and w from {0.35, 0.4, 0.45, 0.5, 0.55}. For all components of the model, rectified linear-unit (ReLU) [77] was used as the activation for improving the computational efficiency and retaining the gradient. Moreover, the Adam optimizer was employed to minimize the loss function through updating network weights and learning rate based on calculated gradients [78].

### 3.4. Evaluation Metrics

Generally, for objectively evaluating the performance of proposed models, the popular metrics including sensitivity (Sn), specificity (Sp), accuracy (Acc) and Matthews correlation coefficient (MCC) [62,79,80,81,82,83,84,85] were used, as shown in Equation (3).
(3)$$\{\begin{array}{c} Sn=\frac{TP}{TP+FN}\;\;\;\;0\leq Sn\leq 1\\ Sp=\frac{TN}{TN+FP}\;\;\;\;0\leq Sp\leq 1\\ Acc=\frac{TP+TN}{TP+TN+FP+FN}\;\;\;\;\;\;\;\;\;\;0\leq Acc\leq 1\\ MCC=\frac{TP\times TN-FP\times FN}{\sqrt{\left(TP+FN\right)\times \left(TN+FN\right)\times \left(TP+FP\right)\times \left(TN+FP\right)}}-1\leq MCC\leq 1 \end{array}$$
where *TP* stands for the number of true m^6^A modification sites correctly predicted; *TN*, the number of true non-m^6^A modification sites correctly predicted; *FP*, the number of non-m^6^A modification sites incorrectly predicted as m^6^A modification sites; and *FN*, the number of m^6^A modification sites incorrectly predicted as non-m^6^A modification sites.

Furthermore, the area under the receiver operating characteristic curve or AUROC [78,86,87] is able to visually display the prediction performance of the proposed models. The value of AUROC ranges from 0 and 1. The larger the value is, the better the prediction performance is.

## 4. Conclusions

As an important RNA transcriptional modification, m^6^A is of vital importance in mRNA translation, mRNA stability, directional differentiation of hematopoietic stem cells and spermatogenesis. Therefore, the ability to accurately identify m^6^A modification sites in a genome at the single-nucleotide level would have profound effects on revealing its regulatory mechanism and assisting drug development. In the current study, we collected high confidence m^6^A modification sites identified by m^6^A-REF-seq from different tissues of three types of mammals, respectively, and constructed 11 tissue-specific models using ensemble deep learning. Our predictors could produce AUROC values in the range of 0.82–0.91 on the corresponding independent testing datasets. Compared with the existing predictors in this field, the AUROC values are improved to a certain degree, indicating that our models have the ability to learn general methylation rules on RNA bases and generalize to m^6^A transcriptome-wide identification. During cross-species/tissues validation, when testing the tissue-specific models trained on the human (brain) and mouse (liver), almost all the achieved AUROC values were less than 0.82, demonstrating that constructing tissue-specific predictors is helpful for better detection of m^6^A sites.

Despite the recent progress on m^6^A site identification in multiple tissues of mammals, there are several limitations outstanding: (1) These predictors for identifying m^6^A sites in multiple tissues of mammals were trained on such m^6^A modification data with a specific ACA motif, causing them difficulty in accurately identifying those m^6^A sites in another motif, such as canonical m^6^A motif DRACH (D = A, G or U; R = A or G; H = C or U). (2) Current computational methods for identifying m^6^A sites in multiple tissues of mammals are limited by algorithmic constraints because these methods were built upon classical machine learning or deep learning methods, such as SVM, CNN, or ensemble deep learning. It is necessary to adopt novel methods to improve the prediction performance of the model in the future. Thus, how to design a prediction method that can comprehensively and accurately predict m^6^A sites across various tissues of mammals is still a challenge.

To this end, we provide several insights for future directions of m^6^A site identification. Firstly, to identify potential m^6^A sites across various tissues of mammals efficiently, high-confidence m^6^A sites that have been experimentally annotated as methylated sites with the canonical m^6^A motif DRACH in tissues of mammals, and those sites that are clearly validated by experiment yet fail to be methylated, are the ideal data source for constructing accurate prediction models. Secondly, the use of graph neural network is also a promising direction in improving the performance of m^6^A modification site prediction. A given sequence can be transformed into a directed pattern graph, where each vertex could be represented by a k-mer of the given sequence. An edge connects two vertices if the joint subsequence of corresponding k-mers appear at least once in this sequence. Our expectation is that within the next few years, m^6^A modification sites will be able to be predicted with a high degree of accuracy, purely based on their sequence.

## Figures and Tables

**Figure 1 ijms-23-15490-f001:**
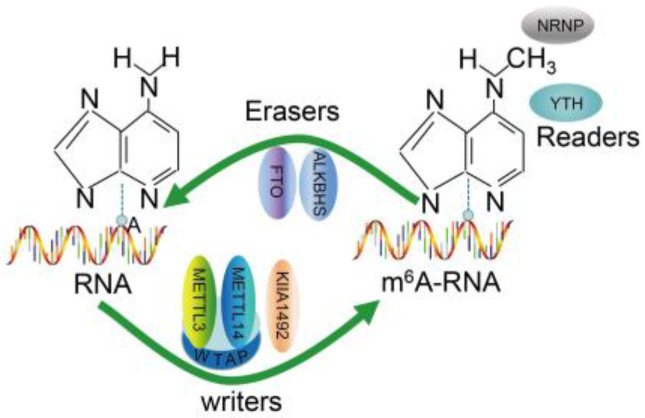
An illustration to show m^6^A modification in RNA. Writer, eraser and reader proteins are thought to install, remove and read m^6^A modifications in RNA, respectively.

**Figure 2 ijms-23-15490-f002:**
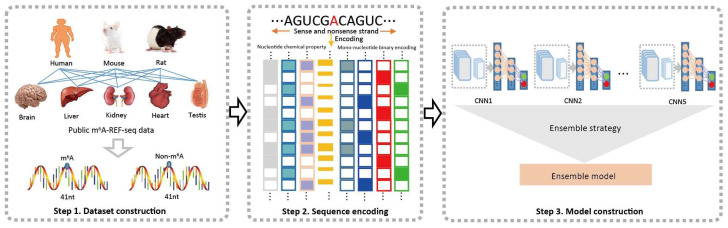
Schematic illustration of the im6APred framework, including three important modules: dataset construction, sequence encoding and model construction.

**Figure 3 ijms-23-15490-f003:**
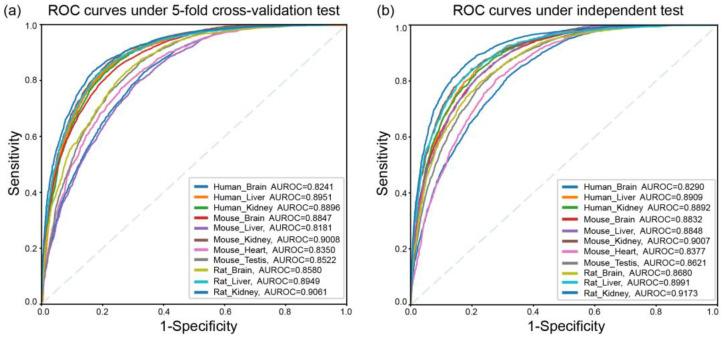
The ROC curves of im6APred: (**a**) the ROC curves of im6APred for identifying m^6^A modification sites in different tissues from the three species under the 5-fold cross-validation test; (**b**) the ROC curves of im6APred under the independent dataset test.

**Figure 4 ijms-23-15490-f004:**
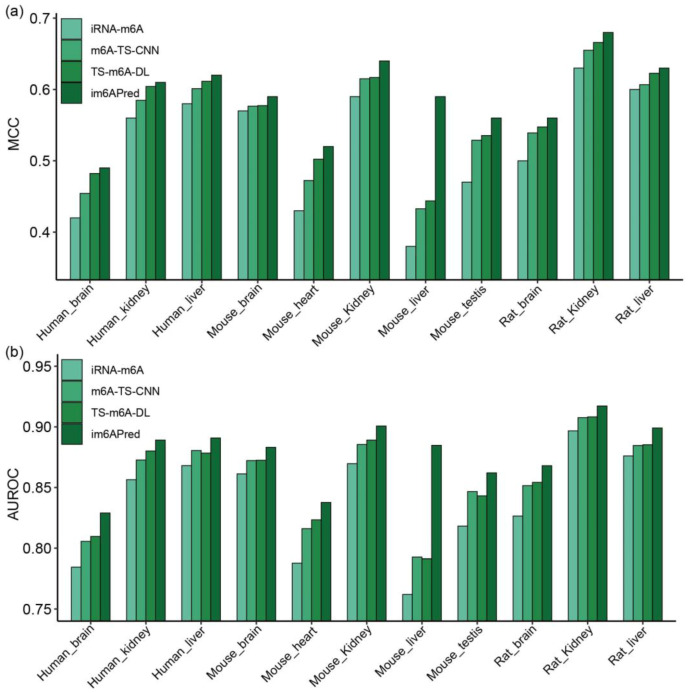
Comparison of the models using an independent test; iRNA-m6A, m6A-TS-CNN, TS-m6A-DL and im6APred in term of MCC and AUROC. (**a**) the MCC values of iRNA-m6A, m6A-TS-CNN, TS-m6A-DL and im6APred; (**b**) the AUROC values of iRNA-m6A, m6A-TS-CNN, TS-m6A-DL and im6APred.

**Figure 5 ijms-23-15490-f005:**
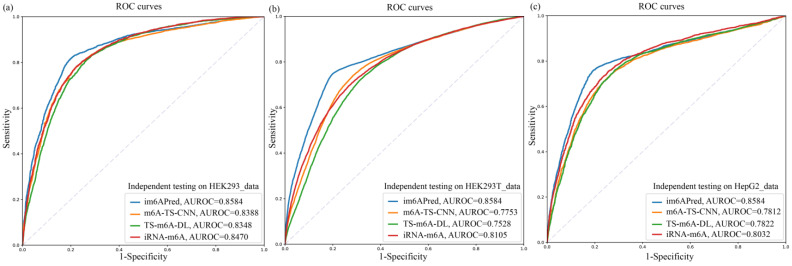
The ROC curves of the tissue-specific models for m^6^A site prediction using three other independent testing sets: (**a**) the ROC curves of the tissue-specific models for m^6^A site prediction using HEK293_data; (**b**) the ROC curves of the tissue-specific models for m^6^A site prediction using HEK293T_data; (**c**) the ROC curves of the tissue-specific models for m^6^A site prediction using HepG2_data.

**Figure 6 ijms-23-15490-f006:**
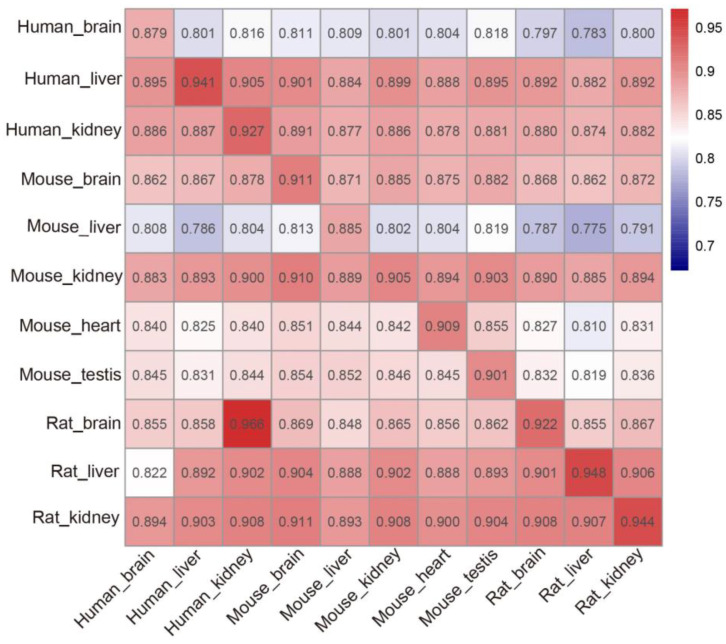
The heat map showing the cross- and intra-tissue prediction AUROC values. Once a tissue-specific model was established on its own training dataset in rows, it was validated on the data from the same tissue as well as the independent data from the other tissues of the three species in columns.

**Table 1 ijms-23-15490-t001:** Characteristics of the existing approaches for RNA m^6^A modification site prediction.

Species	Tool	Experimental Method	Single Nucleotide Resolution	Features ^a^	Algorithm	Evaluation Strategy	Year	Webserver ^b^
*S. cerevisiae*	iRNA-Methyl [24]	m^6^A-Seq	No	PseDNC	SVM	Jackknife	2015	http://lin-group.cn/server/iRNA-Methyl (accessed on 3 December 2022)
	pRNAm_PC [35]	m^6^A-Seq	No	PseDNC, AC, CC	SVM	Jackknife	2015	http://www.jci-bioinfo.cn/pRNAm-PC (accessed on 3 December 2022)
	RNA-MethylPred [43]	m^6^A-Seq	No	DNC, KNN scores	SVM	Jackknife	2016	No
	TargetM6A [36]	m^6^A-Seq	No	PSNP, PSDP, NC	SVM	Jackknife, independent test	2016	http://202.119.84.36:3079/TargetM6A/ (accessed on 3 December 2022)
	RAM-ESVM [30]	m^6^A-Seq	No	PseDNC	Ensemble SVM	10-fold CV	2017	decommissioned
	iRNA(m6A)-PseDNC [29]	m^6^A-Seq	No	PseDNC	SVM	10-fold CV	2018	http://lin-group.cn/server/iRNA(m6A)-PseDNC.php (accessed on 3 December 2022)
	M6APred-EL [34]	m^6^A-Seq	No	PS(k-mer)NP, RFHC-GACs, PCPs	Ensemble SVM	10-fold CV	2018	decommissioned
	DeepM6APred [28]	m^6^A-Seq	No	Deep features, NPPS	SVM	10-fold CV	2018	decommissioned
	iMethyl-STTNC [38]	m^6^A-Seq	No	PseDNC, PseTNC, STNC, STTNC	SVM	10-fold CV	2018	No
*H. sapiens*	iRNA-PseColl [27]	m^6^A-Seq	No	CPD	SVM	Jackknife	2017	http://lin-group.cn/server/iRNA-PseColl.html (accessed on 3 December 2022)
	HMpre [40]	miCLIP	Yes	SLRF, FREI, SNP	XGBoost	Independent test	2018	No
	MultiRM [42]	m6A-CLIP, miCLIP	Yes	One-hot	CNN + BiLSTM	Independent test	2021	www.xjtlu.edu.cn/biologicalsciences/multirm (accessed on 3 December 2022)
	DeepM6ASeq-EL [44]	m6A-CLIP, miCLIP	Yes	One-hot, CPD, Word2vec	Ensemble CNN + LSTM	Independent test	2021	No
*H. sapiens*, *M. musculus*	MethyRNA [26]	m^6^A-Seq, MeRIP-Seq	No	CPD	SVM	Jackknife	2016	http://lin-group.cn/server/MethyRNA (accessed on 3 December 2022)
	SRAMP [39]	miCLIP	Yes	One-hot, SPE, KNN scores, PSSP	RF	5-fold CV, independent test	2016	http://www.cuilab.cn/sramp/ (accessed on 3 December 2022)
	RNAMethPre [32]	miCLIP, m6A-CLIP	Yes	One-hot, NC, SLS	SVM	5-fold CV, independent test	2016	decommissioned
	iRNA-3typeA [33]	m^6^A-Seq, MeRIP-Seq	No	CPD	SVM	Jackknife	2018	http://lin-group.cn/server/iRNA-3typeA.php (accessed on 3 December 2022)
*A. thaliana*	M6ATH [25]	m6A-seq	No	CPD	SVM	Jackknife	2016	http://lin-group.cn/server/M6ATH (accessed on 3 December 2022)
*H. sapiens, Mouse, Zebrafish*	DeepM6ASeq [45]	miCLIP	Yes	One-hot	CNN + BiLSTM	independent test	2018	https://github.com/rreybeyb/DeepM6ASeq (accessed on 3 December 2022)
*S. cerevisiae*, *H. sapiens*, *A. thaliana*	RAM-NPPS [31]	m^6^A-Seq, PA-m6A-seq	No	NPPS	SVM	10-fold CV	2017	decommissioned
*H. sapiens*, *Mouse*, *Chimpanzee*, *Rhesus*, *Pig*, *Rat*, *Zebrafish*	MASS [41]	m6A-Seq, MeRIP-Seq, m6A-CLIP, miCLIP	Bulking	One-hot, Phylogenetic tree	CNN + BiLSTM	5-fold CV	2021	https://github.com/mlcb-thu/MASS (accessed on 3 December 2022)
*H. sapiens*, *M. musculus*, *Rat*	iRNA-m6A [46]	m6A-REF-seq	Yes	PCPs, CPD, One-hot	SVM	5-fold CV, independent test	2020	http://lin-group.cn/server/iRNA-m6A/ (accessed on 3 December 2022)
	im6A-TS-CNN [47]	m6A-REF-seq	Yes	One-hot	CNN	5-fold CV, independent test	2021	No
	TS-m6A-DL [48]	m6A-REF-seq	Yes	One-hot	CNN	5-fold CV, independent test	2021	http://nsclbio.jbnu.ac.kr/tools/TS-m6A-DL/ (accessed on 3 December 2022)

^a^ PseDNC: pseudo dinucleotide composition; DNC: dinucleotide composition; AC: auto-covariance; CC: cross-covariance; KNN scores: K-nearest neighbor encoding; PSNP: position-specific nucleotide propensity; PSDP: position-specific dinucleotide propensity; NC: nucleotide composition; PS(k-mer)NP: position-specific k-mer nucleotide propensity; PCPs: physical-chemical properties; CPD: chemical property with density, RFHC-GAC: a method integrating by CPD, AC and CC; NPPS: nucleotide pair position specificity; PseTNC: pseudo-trinucleotide-composition; STNC: split-trinucleotide-composition; STTNC: split-tetranucleotide-composition; PSSP: predicted secondary structure pattern; spectrum encoding: nucleotide pair spectrum encoding; SLS: stability of the local structure; one-hot: binary encoding; SPE: spectrum encoding, SLRF: site location related features, FREI: features related to entropy information, SNP: single nucleotide polymorphism features. ^b^ decommissioned—the webserver/tool is no longer available; no—the publication has no webserver or tool.

**Table 2 ijms-23-15490-t002:** Performance comparison between different base–classifier combinations on human brain training data using five-fold cross validation.

Ensemble Framework	Base–Classifier Combination	ACC (%)	Sn (%)	Sp (%)	MCC	AUROC
Single classifier	(1) SVM	70.79	74.25	67.34	0.42	0.7746
	(2) KNN	67.97	71.10	64.84	0.36	0.7526
	(3) RF	70.80	76.37	65.23	0.42	0.7776
	(4) GB	70.74	75.66	65.82	0.42	0.7782
	(5) CNN	72.94	77.59	68.30	0.46	0.8139
	(6) FCN	69.21	70.29	68.12	0.38	0.7479
	(7) LSTM	72.67	78.05	67.30	0.46	0.8047
Ensemble across multiple models	(1) + (2) + (3) + (4) + (5)	71.64	75.94	67.34	0.43	0.8028
	(1) + (2) + (3) + (4) + (6)	70.92	75.70	66.15	0.42	0.7763
	(1) + (2) + (3) + (4) + (7)	71.53	76.50	66.56	0.43	0.8071
	(1) + (2) + (3) + (5) + (6)	71.35	76.92	65.78	0.43	0.7989
	(1) + (2) + (4) + (5) + (6)	71.18	75.70	66.67	0.43	0.7986
	(1) + (3) + (4) + (5) + (6)	71.90	76.66	67.14	0.44	0.8000
	(2) + (3) + (4) + (5) + (6)	71.66	75.68	67.64	0.43	0.8029
	(1) + (2) + (3) + (6) + (7)	71.27	76.35	66.19	0.43	0.8027
	(1) + (2) + (4) + (6) + (7)	70.98	75.33	66.62	0.42	0.8009
	(1) + (3) + (4) + (6) + (7)	71.66	77.37	65.95	0.44	0.8026
	(1) + (2) + (3) + (5) + (7)	72.38	78.35	66.41	0.45	0.8172
	(1) + (2) + (4) + (5) + (7)	72.42	77.42	67.43	0.45	0.8150
	(1) + (3) + (4) + (5) + (7)	72.81	79.91	65.71	0.46	0.8165
	(2) + (3) + (4) + (5) + (7)	72.55	77.29	67.82	0.45	0.8188
	(1) + (2) + (5) + (6) + (7)	72.10	74.77	69.42	0.44	0.8137
	(2) + (3) + (4) + (6) + (7)	71.57	77.98	65.17	0.44	0.8050
	(1) + (3) + (5) + (6) + (7)	72.48	77.02	67.93	0.45	0.8129
	(1) + (4) + (5) + (6) + (7)	72.30	76.96	67.64	0.45	0.8130
	(2) + (3) + (5) + (6) + (7)	72.20	77.29	67.12	0.45	0.8170
	(2) + (4) + (5) + (6) + (7)	72.40	77.09	67.71	0.45	0.8142
	(3) + (4) + (5) + (6) + (7)	72.38	78.09	66.67	0.45	0.8135
Ensemble with same kind of model	(1) + (1) + (1) + (1) + (1)	70.42	73.88	66.97	0.41	0.7718
	(2) + (2) + (2) + (2) + (2)	68.58	71.62	65.54	0.37	0.7564
	(3) + (3) + (3) + (3) + (3)	70.93	76.74	65.12	0.42	0.7793
	(4) + (4) + (4) + (4) + (4)	70.81	75.74	65.88	0.42	0.7773
	(5) + (5) + (5) + (5) + (5)	74.23	82.48	65.99	0.49	0.8241
	(6) + (6) + (6) + (6) + (6)	69.71	69.45	69.97	0.39	0.7749
	(7) + (7) + (7) + (7) + (7)	73.79	81.04	66.54	0.48	0.8200

**Table 3 ijms-23-15490-t003:** Performance of im6APred under the 5-fold cross-validation test and an independent test.

Species	Tissues	Five-Fold Cross Validation					Independent Test				
		ACC (%)	Sn (%)	Sp (%)	MCC	AUROC	ACC (%)	Sn (%)	Sp (%)	MCC	AUROC
Human	Brain	74.23	82.48	65.99	0.49	0.8241	74.38	80.41	68.35	0.49	0.8290
	Liver	81.55	84.13	78.98	0.63	0.8915	81.21	82.61	79.80	0.62	0.8909
	Kidney	80.67	84.89	76.45	0.62	0.8896	80.59	87.29	73.89	0.61	0.8892
Mouse	Brain	79.85	83.54	76.16	0.60	0.8847	79.50	83.18	75.83	0.59	0.8832
	Liver	73.53	84.01	63.05	0.48	0.8181	78.61	91.34	65.88	0.59	0.8848
	Kidney	81.96	83.51	80.42	0.64	0.9008	81.78	81.96	81.60	0.64	0.9007
	Heart	75.31	81.69	68.92	0.51	0.8350	75.91	82.55	69.27	0.52	0.8377
	Testis	76.90	85.70	68.09	0.54	0.8522	77.61	84.49	70.74	0.56	0.8621
Rat	Brain	77.27	81.80	72.75	0.55	0.8580	77.20	87.32	67.08	0.56	0.8680
	Liver	81.81	82.24	81.38	0.64	0.8949	81.44	86.66	76.22	0.63	0.8991
	Kidney	82.97	82.90	83.05	0.66	0.9061	83.92	84.91	82.93	0.68	0.9173

**Table 4 ijms-23-15490-t004:** Performance evaluation of the tissue-specific models for m^6^A site prediction using three other independent testing sets.

Independent Test Dataset	Tissues	Predictors	Evaluation Index				
			ACC (%)	Sn (%)	Sp (%)	MCC	AUROC
HEK293_data	Kidney	iRNA-m6A	77.80	78.46	77.13	0.5560	0.8470
		m6A-TS-CNN	79.32	75.99	82.65	0.5877	0.8388
		TS-m6A-DL	77.45	72.55	82.34	0.5517	0.8348
		im6APred	81.36	82.93	79.78	0.6275	0.8584
HEK293T_data	Kidney	iRNA-m6A	70.84	63.23	78.45	0.4217	0.7751
		m6A-TS-CNN	74.80	68.41	81.18	0.5001	0.7753
		TS-m6A-DL	72.66	62.62	82.70	0.4627	0.7528
		im6APred	78.38	76.03	80.73	0.5682	0.8105
HepG2_data	Liver	iRNA-m6A	74.80	73.20	76.39	0.4962	0.8032
		m6A-TS-CNN	77.01	72.98	81.03	0.5419	0.7812
		TS-m6A-DL	76.21	75.95	76.48	0.5243	0.7822
		im6APred	78.92	76.97	80.87	0.5789	0.8113

**Table 5 ijms-23-15490-t005:** The benchmark datasets for identifying RNA m^6^A modification sites.

Species	Tissues	Positive		Negative	
		Training	Testing	Training	Testing
Human	Brain	4605	4604	4605	4604
	Liver	2634	2634	2634	2634
	Kidney	4574	4573	4574	4573
Mouse	Brain	8025	8025	8025	8025
	Liver	4133	4133	4133	4133
	Kidney	3953	3952	3953	3952
	Heart	2201	2200	2201	2200
	Testis	4707	4706	4707	4706
Rat	Brain	2352	2351	2352	2351
	Liver	1762	1762	1762	1762
	Kidney	3433	3432	3433	3432

## Data Availability

The training and independent testing used in this study can be downloaded from http://47.94.248.117/im6APred/download (accessed on 3 December 2022) or https://github.com/pythonLzt/im6APred (accessed on 3 December 2022).

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
