# Peer review of "Predicting N6-Methyladenosine Sites in Multiple Tissues of Mammals through Ensemble Deep Learning"

_ijms, 2022, doi:10.3390/ijms232415490_

Round 1

Reviewer 1 Report

This manuscript by Luo and coworkers established a new M6A modification prediction website server, using the ensemble deep learning method. I think the novelty is very low for this paper. From Figure 4, we can see the method is only very slightly better than the existing prediction tool, especially comparing to the TS-m6A-DL.

The authors should  strengthen more on the performance of the new tool and provide more compelling data about the improved prediction accuracy of this tool, and discuss the real advancement of this tool compared to the existing ones.

Reviewer 2 Report

ijms-2043144

In this manuscript, Luo et al are describing the tool they have developed by ensemble deep learning-based method. This tool can predict m6A modification on primary RNA molecules from different tissue samples. Overall, the manuscript is well-described and can be considered for publication. I have few points that need to be addressed before publication.

·      In cross-validation, I am concerned about how accuracy, sensitivity, specificity, and MCC are calculated. As SVM outputs a score for each testing dataset, which score cutoff is used to define TF, FP?

·      In the independent testing dataset which score cutoff is used to define TF, FP?

·      Author did not mention how they select the datasets. Also, explain the procedure used in dataset cleaning. Kindly provide the exact data used to allow better reproducibility of the research. The authors should incorporate all the information regarding the dataset and the analysis in a way that helps reproducibility.

·      Author also needs to provide their code which allows better reproducibility of the research and helps researchers working in this field. I wonder which kernel function was used for each of the predictions.

·      Author performs 5-fold cross-validation and uses 80% of the data as a training set and 20% as an independent testing set. Are the authors divide the dataset into training and independent testing and then separately perform the analysis or do they randomly divide the dataset during the analysis? Please make it clear.

·      In the benchmark dataset, the author mentioned, they removed the sequences with more than 80% sequence similarity by using CD-HIT. The author needs to make it clear why they used 80% sequence similarity. I would like to see the comparative performance at 60% and 40%.

·      Is the CD-HIT the only method to remove similar sequences? If not, please make clear why the authors use this method.

·      Author used SVM, KNN, RF, GB, FCN, LSTM, and CNN classifiers however they did not mention how they trained the model and what parameter they used to train it. Please incorporate the parameter in the supplementary table.

·      In Table 1, please incorporate the URL in a separate column, not in table legends.

·      In terms of MCC and AUROC, both TS-m6A-DNN and im6Apred have nearly similar values. The authors need to elaborate on how the tool (im6Apred) developed here is significantly better than the already existing tool like TS-m6A-DNN.

·      Reference numbers in table 1 are not matching with the list of references in the reference section.

·      If the user submits a plant protein to the server, what output user will get?

·      Paper contribution and work challenges should be clear enough for readers to provide information about the major conclusion, limitations, and future directions to the present study. 

Round 2

Reviewer 1 Report

All points are addressed well.